# Development of a Novel Scoring Model to Estimate the Severity Grade of Mesenteric Artery Stenosis

**DOI:** 10.3390/jcm11247420

**Published:** 2022-12-14

**Authors:** Safwan Omran, Frank Konietschke, Verena Mueller, Maximilian de Bucourt, Jan Paul Frese, Andreas Greiner

**Affiliations:** 1Department of Vascular Surgery, Charité—Universitätsmedizin Berlin, Corporate Member of Freie Universität Berlin and Humboldt-Universität zu Berlin, 12203 Berlin, Germany; 2Institute of Medical Biometrics and Clinical Epidemiology, Charité—Universitätsmedizin Berlin, Corporate Member of Freie Universität Berlin and Humboldt-Universität zu Berlin, 10117 Berlin, Germany; 3Berlin Institute of Health (BIH), Anna-Louisa-Karsch-Straße 2, 10178 Berlin, Germany; 4Department of Radiology, Campus Benjamin Franklin, Charité—Universitätsmedizin Berlin, Corporate Member of Freie Universität Berlin and Humboldt-Universität zu Berlin, 12203 Berlin, Germany

**Keywords:** chronic mesenteric ischemia, mesenteric artery stenosis, celiac artery, superior mesenteric artery, scoring model

## Abstract

Objective: This study aimed to derive a new scoring model from estimating the severity grade of mesenteric artery stenosis. We sought to analyze the relationship between the new scoring model and the development, treatment, and mortality of chronic mesenteric ischemia (CMI). Methods: This retrospective study included 242 patients (128 (53%) women and 114 (47%) men) with suspected CMI from January 2011 to December 2020. A weighted sum six-point score (CSI-score; the celiac artery is abbreviated by “C”, superior mesenteric artery by “S”, and inferior mesenteric artery by “I”) based on the number of affected vessels and the extent and grade of the stenosis or occlusion of the involved visceral arteries was derived by maximizing the area under the ROC curve. The calculated CSI-score ranged from 0 to 22. The patients were divided according to the best cut-off point into low-score (CSI-score < 8) and high-score (CSI-score ≥ 8) groups. Results: The area under the receiver operating characteristic curve (AUC) of the CSI-score was 0.86 (95% CI, 0.82–0.91). The best cut-off point of “8” represented the highest value of Youden’s index (0.58) with a sensitivity of 87% and specificity of 72%. The cohort was divided according to the cut-off point into a low-score group (*n* = 100 patients, 41%) and high-score group (*n* = 142 patients, 59%) and according to the clinical presentation into a CMI group (*n* = 109 patients, 45%) and non-CMI group (*n* = 133 patients, 55%). The median CSI-score for all patients was 10 (range: 0 -22). High-scoring patients showed statistically significant higher rates of coronary artery disease (54% vs. 36%, *p* = 0.007), chronic renal insufficiency (50% vs. 30%, *p* = 0.002), and peripheral arterial disease (57% vs. 16%, *p* < 0.001). A total of 109 (45%) patients underwent invasive treatment of the visceral arteries and were more often in the high-score group (69% vs. 11%, *p* < 0.001). Of those, 79 (72%) patients underwent primary endovascular treatment, and 44 (40%) patients underwent primary open surgery or open conversion after endovascular treatment. Sixteen (7%) patients died during the follow-up, with a statistically significant difference between high- and low-scoring patients (9% vs. 0%, *p* = 0.008). The score stratification showed that the percentage of patients treated with endovascular and open surgical methods, the recurrence of the stenosis or failure of the endovascular treatment, the need for a bypass procedure, and the mortality rates significantly increased in the subgroups. The CSI-score demonstrated an excellent ability to discriminate between patients who needed treatment and those who did not, with an AUC of 0.87 (95% CI, 0.82–0.91). Additionally, the CSI-score’s ability to predict the patients’ mortality was moderate, with an AUC of 0.73 (95% CI, 0.62–0.83). Conclusions: The new scoring model can estimate the severity grade of the stenosis of the mesenteric arteries. Our study showed a strong association of the score with the presence of chronic mesenteric ischemia, the need for treatment, the need for open surgery, and mortality.

## 1. Introduction

Chronic mesenteric ischemia (CMI) is a disorder caused by chronic impairment of visceral arterial blood flow [1]. The most common cause of CMI is the atherosclerotic occlusive disease of the origin of the mesenteric vessels [2]. It has already been established that mesenteric arterial stenosis is relatively common, occurring in about 17.5% of elderly patients [3]. On the other hand, the incidence of CMI is low due to the collateral flow in the celiac–mesenteric arterial system.

CMI is widely described as an association of postprandial pain, significant weight loss, and unspecific gastrointestinal symptoms. However, its relationship to atherosclerotic lesions of the mesenteric arteries is still unclear. It is well known that some patients with massive mesenteric atherosclerosis can remain asymptomatic for a long time. On the other hand, many other patients with single-vessel stenosis can develop symptoms of chronic mesenteric ischemia.

The current study aims to develop a new scoring model to estimate the severity grade of the stenosis of the mesenteric arteries and its relationship to the development of chronic mesenteric ischemia and to guide the treatment of CMI patients.

## 2. Materials and Methods

The Hospital Ethics Committee approved the study and waived the patients’ need for informed consent (Number: EA4/009/19).

### 2.1. Study Design, Setting, and Participants

For this study, the German adaptation of the International Classification of Diseases (ICD-10-GM) was used to identify patients admitted with chronic mesenteric ischemia (ICD-10-GM code K55.1). A total of 331 patients with the code K55.1 were identified retrospectively from January 2011 to December 2020. Exclusion criteria included active cancers diagnosed within the previous six months and recurrent, regionally advanced, or metastatic cancers. Additionally, patients <18 years old, patients with acute embolic occlusion of the mesenteric arteries, and patients who did not have imaging studies of the visceral arteries were excluded (see Figure 1). As a result, 242 patients with suspected CMI (with the ICD-10-GM code K55.1) achieved the inclusion criteria and were enrolled in the current study. All these patients had previously been referred for outpatient treatment with suspicion of CMI. However, the patients underwent further diagnostics and multidisciplinary consensus to confirm or exclude the diagnosis of CMI. Thus, patients deemed to have CMI were classified into the CMI group and those who did not in the non-CMI group. Criteria for distinguishing patients with CMI from those without CMI were based on symptoms at presentation, onset time, duration of symptoms, laboratory findings, endoscopic findings, CT, and angiographic imaging suggestive of ischemia. The symptoms under consideration included postprandial abdominal pain, unintentional weight loss, nausea, diarrhea, and fear of eating.

All individuals had undergone an abdominal computed tomography angiography (CTA) using multidetector computed tomography (MDCT). To develop and calculate the new score, we retrospectively analyzed the CTA of all patients using a set of advanced postprocessing techniques and quantitative vessel analysis (QVA), including multiplanar reconstruction (MPR), curved planar reformation (CPR), and volume rendering (VR). If the measurement of the stenosis grade was not otherwise suitable, centerline analysis was performed. Two experienced surgeons independently evaluated the visceral arteries of all patients and calculated the new score for every patient based on the measurements on the CTA images. Measurements of the stenosis and calculating the score were entirely independent of the categorization according to CMI and were not used for multidisciplinary consensus decision making during the study period.

### 2.2. Definitions

In the current study, CMI was defined as insufficient blood supply to the gastrointestinal tract leading to ischemic symptoms with a duration of at least three months [4].

In addition, postprandial abdominal pain was defined as abdominal pain occurring within 15 to 30 min after eating. Moreover, endovascular treatment failure was defined as failed recanalization of the target visceral vessels, and recurrent stenosis after endovascular treatment was defined as the development of restenosis after technically and clinically successful recanalization of a visceral artery.

### 2.3. The New Classification

Visceral artery stenosis was determined based on CT angiography analysis according to the extent of stenosis (ostial, proximal, and extended stenosis) and the grade of stenosis (0–50%, 51–70%, 71–99%, and occlusion). Ostial stenosis was defined as stenosis of the visceral artery within the first 10 mm from the aortic lumen. The proximal lesion represents stenosis into the arterial segment extending from the aortic lumen to the origin of the inferior pancreaticoduodenal artery (IPDA), the first branch of the superior mesenteric artery (SMA), and from the aortic lumen to the bifurcation of the celiac artery (CA). Extended stenosis was defined as stenosis extending beyond the SMA’s first branch and the celiac artery’s bifurcation. Only ostial stenosis was considered regarding the inferior mesenteric artery (IMA).

The new classification attributes a scale from 0 to 3 for the extent of the lesions (none, ostial, proximal, and extended stenosis) and the maximum stenosis grade (0–50%, 51–70%, 71–99%, and occlusion), as described in Figure 2.

The grade of stenosis was measured in terms of the residual lumen’s diameter (D stenosis) and the estimated normal luminal diameter at the level of the stenosis (D normal) [5]. Therefore, the percentage of stenosis was calculated as (1 − D stenosis/D normal) × 100 (see Figure 3). In the case of generalized calcification of the artery, we estimated the degree of stenosis in terms of the narrowest diameter.

This classification describes the lesions in a normal anatomical pattern, representing about 75% of the population [6,7]. However, in most cases, the absence of one of the visceral arteries and other anatomical variations could be considered in this classification.

### 2.4. Development of the CSI-Score and Endpoints

For the development of the CSI-score, the first step was to define the values of each of the primary variables for each patient. The primary six variables (C_extent_, C_grade_, S_extent_, S_grade_, I_extent_, I_grade_) represent the scales of the extent and grade of the stenosis of each visceral artery taken from the CTA (see Figure 2). CA is abbreviated by “C”, SMA by “S”, and IMA by “I”. We suggest combining these variables into a single score. In order to find the best equation that discriminates between CMI and non-CMI patients, we performed a grid search on the set of parameters. We found that the following equation of the score yielded the highest AUC of 0.86 (95% CI, 0.82–0.91).
CSI-score = 1 × C_extent_ + 2 × C_grade_ + 1 × S_extent_ + 2 × S_grade_ + 1 × I_extent_ + 1 × I_grade_

Finally, the CSI-score was calculated for every patient according to this equation. The calculated CSI-score ranged from 0 to 22. The optimum cut-off point to discriminate between CMI and non-CMI patients was 8. This cut-off point presented the highest value of Youden’s index (0.58) with a sensitivity of 87% and specificity of 72%. After that, the patients were divided into two groups: low score (<8) and high score (≥8).

The primary endpoint of this study was the association of CSI-score with chronic mesenteric ischemia. Secondary endpoints were mortality, the need for treatment, endovascular failure or recurrence of the stenosis after endovascular treatment, and the need for bypass procedures.

### 2.5. Bootstrap Analysis and Score Validation

Due to the lack of a validation set or external validation, it is not possible to report score prediction values at this stage. However, to assess its initial predictive quality, we assessed its discriminative quality in a bootstrap analysis. We generated a random data set by drawing with replacement from the data set (of the same size), calculated the CSI-score, and estimated the area under the ROC curve. We repeated these steps 100,000 times and ended up with a minimum AUC of 75% and an average AUC of 87% with a 95% confidence interval of [82%, 91%].

### 2.6. Statistical Methods

The diagnostic accuracies were obtained in terms of the areas under the ROC curves (AUCs) with nonparametric rank-based estimators. Furthermore, for the computation of 95% confidence intervals, we inverted the Brunner–Munzel test. Additionally, we used standard methods (depending on scales) for data descriptions. Finally, the inter-rater reliability was determined by percent agreement and the kappa statistic. All computations and data evaluations were performed using SPSS 25 (IBM, Armonk, NY, USA) and R statistical computing (R-4.2.2 for Windows) software (R Core Team (2020). R: A language and environment for statistical computing. R Foundation for Statistical Computing, Vienna, Austria. URL https://www.R-project.org/).

## 3. Results

### 3.1. Study Populations

Of the 331 recruited patients, a total of 89 were excluded for the reasons shown in (Figure 1). As a result, 242 patients (128 (53%) women and 114 (47%) men) were enrolled in the study. According to the definitions and criteria mentioned above, the cohort was divided into the CMI group (*n* = 109) and the non-CMI group (*n* = 133). The ultimate diagnosis of non-CMI patients included chronic gastritis in 34 (26%) patients, ischemic colitis in 23 (17%), peptic ulcer in 9 (7%), diverticulitis in 7 (5%), small-bowel obstruction in 3 (2%), and psychological reasons in 9 (7%) patients. The remaining 48 (36%) patients did not have a definitive diagnosis and were discharged home.

Of the patients included in this study, 196 (81%) patients had visceral artery stenosis, and 46 (19%) patients had no stenosis. In terms of etiology, atherosclerosis was the most common cause of stenosis, in 185 (94%) patients, and median arcuate ligament (MALS) syndrome was found in 11 (6%) patients.

All CMI patients had stenosis of at least one of the visceral arteries, including 13 (12%) with single-vessel stenosis, 32 (29%) with two-vessel stenosis, and 64 (59%) with three-vessel stenosis. On the other hand, 90 (68%) of the non-CMI patients had stenosis of the visceral arteries, including 42 (32%) with single-vessel stenosis, 21 (16%) with two-vessel stenosis, and 27 (20%) with three-vessel stenosis. According to the stenotic vessels in the CMI group, celiac artery stenosis was found in 98 (90%) patients, SMA stenosis in 99 (91%) patients, and IMA stenosis in 72 (66%) patients.

The new CSI-score was calculated for all patients and showed a median value of 10 (range: 0–22). The median value of the score in the CMI group was 14 (range: 5–22) and in the non-CMI group was 4 (range: 0–20) with *p* < 0.001. Overall agreement between the two examiners in calculating the CSI-score occurred in 201 of 242 cases for a percentage agreement of 83%. This resulted in a kappa coefficient of 0.82 (95% confidence interval of 0.77–0.87). The patients were divided according to the best cut-off point (CSI ≥ 8) into two groups: low-score group (*n* = 100, 41%) and high-score group (*n* = 142, 59%).

### 3.2. Risk Factors

CMI was found more often in the high-score group compared to the low-score group (70% vs. 12%, *p* < 0.001). In addition, we observed a similar prevalence of COPD, diabetes mellitus, hypertension, hyperlipoproteinemia, and stroke exacerbations. However, the high-score group showed statistically significant higher rates of coronary artery disease (54% vs. 36%, *p* = 0.007), chronic renal insufficiency (50% vs. 30%, *p* = 0.002), and peripheral arterial disease (57% vs. 16%, *p* < 0.001). Demographics and risk factors are depicted in Table 1.

### 3.3. Clinical Presentation and Manifestations

According to the clinical presentation, abdominal pain was the most frequent symptom, observed in 147 (61%) patients and more often in the high-scoring patients (85% vs. 27%, *p* < 0.001). The classic postprandial pain symptoms were seen in 95 (39%) of the patients and more often in the high-scoring patients (55% vs. 17%, *p* < 0.001). Constant abdominal pain was the main symptom in 52 (22%) patients and was more often in the high-scoring patients (30% vs. 10%, *p* < 0.001). The body mass index was lower in high-scoring patients (23 vs. 25 kg/m^2^, *p* = 0.009). Additionally, the high-scoring patients had more weight loss (38% vs. 13%, *p* < 0.001) and loss of appetite (42% vs. 20%, *p* < 0.001). Further diagnostic investigations, including gastroduodenoscopy, coloscopy, and CT angiography, showed ischemic gastritis in 25 (10%) patients, ischemic hepatitis in 1 (0.4%), and ischemic colitis in 44 (18%) (see Table 2).

### 3.4. Treatment and Outcome

Invasive treatment of the visceral arteries was applied in 109 (45%) patients and was more frequent in the high-score group (69% vs. 11%, *p* < 0.001). A total of 80 (33%) patients underwent primary endovascular treatment, with a higher rate in the high-score group (50% vs. 9%, *p* < 0.001). Primary endovascular success was achieved in 69 (86%) of the patients treated with endovascular methods, including 8 of the 9 (89%) patients in the low-score group and 61 of 71 (86%) in the high-score group. Endovascular treatment failure occurred in 12 (15%) patients, and recurrence of the stenosis after successful endovascular treatment in 10 (13%). The median time between intervention and recurrence was 177 days (range: 43–718 days).

Of the patients who developed recurrence of the stenosis or failure, 17 (7%) underwent open vascular surgical treatment with a higher rate in the high-score group (11% vs. 1%, *p* = 0.002). Open surgical revascularization of the visceral arteries was performed in 44 (18%) patients and more often in the high-score group (29% vs. 3%, *p* < 0.001).

Sixteen (7%) patients died during the follow-up (median: 18 months, range: 1–154 months) with a statistically significant difference between high- and low-scoring patients (11% vs. 1%, *p* = 0.003). The mortality rates showed no statistically significant difference between CMI and non-CMI patients (8% vs. 5%, *p* = 0.351). The causes of death included persistent sepsis and septic shock related to the development of intestine or colonic ischemia resulting in multiple organ failure in eight patients, myocardial infarction in two, advanced liver cirrhosis in one, intracranial hemorrhage in one, and unknown in four.

We analyzed the treatment options and mortality rates in different CSI-score subgroups (see Figure 4). Invasive treatment was not applied in any patient with a CSI-score of <4, and there were no deaths in this subgroup. However, the percentage of patients treated for visceral artery stenosis significantly increased in the subgroups, with a consistent increase in the rate of patients treated with endovascular or open surgical methods. However, the recurrence and failure rates of the endovascular treatment and the need for a bypass procedure significantly increased in the subgroups. While patients with a CSI-score below 4 did not require treatment and did not die during the follow-up period, patients with a CSI-score above 15 had the following results: treatment in 86% (30/35), endovascular treatment in 57% (20/35), endovascular failure in 25% (5/20), recurrent stenosis after endovascular treatment in 15% (3/20), open surgery in 49% (17/35), bypass procedure in 40% (14/35), and mortality of 14% (5/35). The CSI-score demonstrated an excellent ability to discriminate between patients who needed surgical or endovascular treatment due to the symptoms of CMI and those who did not, with an AUC of 0.87 (95% CI, 0.82–0.91). Additionally, the CSI-score’s ability to predict mortality was moderate, with an AUC of 0.73 (95% CI, 0.62–0.83).

## 4. Discussion

A new scoring model was derived from a new classification of the stenosis of the visceral arteries. While the new score included ostial, proximal, and extended stenosis of the CA and SMA, only ostial stenosis of the IMA was included in the score due to the difficulty in evaluating its branches. Additionally, we wanted to avoid the overestimation of the IMA stenosis. The differentiation between ostial and proximal lesions was because ostial lesions are more accessible for endovascular treatment than proximal ones.

All patients in this study received a primary diagnosis of suspected CMI in the outpatient department. However, after further diagnostics and multidisciplinary consensus, 55% of the patients were not found to have CMI. The misdiagnosis rate of CMI is high because the symptoms of CMI are nonspecific and similar to the symptoms of other gastrointestinal diseases [1]. Therefore, CMI is commonly mistaken for other conditions, such as peptic ulcers, gastritis, and diverticulitis. In addition, CTA and MRA imaging manifestations of CMI are variable and often nonspecific. Therefore, diagnosing CMI must follow strict clinical criteria, including the symptoms, time of onset, duration, weight loss, duplex scanning, and endoscopy. Although excluding the CMI diagnosis in patients means that they have sufficient blood supply in the gastrointestinal tract, it does not mean that they have regular visceral arteries without stenosis or occlusions.

While the diagnosis of CMI is based on clinical findings and radiologic manifestations [8,9], the CSI-score is entirely independent of the clinical findings and is a pure morphological classification of the radiologic findings. However, we found a strong association between this score and clinical manifestations, treatment options, and outcomes. The new scoring model demonstrated a high discriminatory ability between patients with CMI and those without. In addition, the score stratification showed that the percentage of patients treated with endovascular and open surgical methods, the recurrence of the stenosis or failure of the endovascular treatment, the need for a bypass procedure, and the mortality rates significantly increased in the subgroups. Based on these results, we not only need to classify the patients into CMI and non-CMI groups but also to correlate them with the score values. In addition, high-scoring non-CMI patients are expected to be more likely to develop symptoms than low-scoring patients. However, this idea needs to be investigated separately in a randomized prospective study. Thomas et al. have suggested that prophylactic mesenteric artery reconstruction should be considered in patients with significant three-vessel mesenteric arterial stenosis [10].

Many studies have reported the classification of visceral arterial stenosis according to the number of stenotic vessels (one, two, or three), the stenotic vessels (CA, SMA, or IMA), or the degree of stenosis [9,11,12,13]. Additionally, several recent studies have compared endovascular treatment with open surgical repair in patients with CMI [14,15,16,17,18,19]. However, there is a lack of literature that describes the selection criteria between endovascular and open surgical treatment. Harki et al. [8] developed a scoring model to predict the risk of chronic gastrointestinal ischemia based on the radiological assessment of the mesenteric arteries and clinical characteristics. Only CA and SMA stenosis and the stenosis grades of <50%, 50% to 70%, and >70% were considered in this study. Other predictors for this scoring model included gender, weight loss, and cardiovascular disease. The results of this score are validated by other studies [9,20]. In contrast to the present study, none of the existing studies reported on the extension of stenosis of the visceral artery as a possible a predictor of CMI and outcome. It is well known that the extension of the stenosis beyond the collateral arteries affects the compensation of the celiac–mesenteric arterial system [21,22].

Duplex scanning is the most used initial imaging study for diagnosing visceral arterial stenosis or occlusion [1,4,23]. In addition, some studies have used duplex scanning to classify the stenosis of CA and SMA [24,25,26]. However, this procedure is technically challenging and requires expertise in ultrasound imaging, visceral arterial hemodynamics, and duplex scan interpretation. On the other hand, CTA with 2D and 3D imaging can provide excellent reconstructions of the visceral arteries and is recommended as the initial imaging study in many studies [4,27,28,29,30,31,32,33,34,35,36,37]. Furthermore, centerline analysis may be helpful in increasing the accuracy of the measurement of the stenosis and is especially appropriate for preoperative planning of endovascular treatment. Moreover, CTA has higher spatial resolution than contrast-enhanced MRA, enabling a more accurate evaluation of peripheral visceral branches [29]. However, MRA has high sensitivity and specificity and allows flow measurement without radiation exposure [38,39,40].

It is generally assumed that symptoms of mesenteric ischemia usually do not appear until at least two of the three visceral arteries are significantly stenosed or occluded [41,42]. However, in the current study, single-vessel stenosis was found in 12% of CMI patients. The onset of symptoms in patients with single-vessel stenosis may be related to the dysfunction of the collateral network of the celiac–mesenteric arterial system. However, in single-vessel stenosis, functional tests are recommended to confirm the diagnosis of CMI [1].

In the present study, high-scoring patients (69%) were more likely to have CMI than low-scoring patients (11%). However, it is well known that single-vessel stenosis of the visceral arteries can cause symptoms of CMI and requires treatment. In contrast, patients with generalized visceral arterial calcification may remain asymptomatic for a long time. Therefore, we cannot assign a specific score value to confirm or exclude a diagnosis of CMI. However, all CMI patients in this study had a score between 5 and 22. In addition, the study population was not randomly selected. Therefore, scoring patients and following them for the development of CMI in a random and larger group of patients will be of great value.

According to the application of the score, the ultimate and detailed values of the CSI-score may help in decision making. Detailed score values indicate the diseased, narrowed blood vessels and the location, severity, and extent of the narrowing. For example, patients with ostial or proximal stenosis of the CA and SMA may benefit from endovascular treatment, and patients with extended stenosis or occlusion of the CA, SMA, or both may benefit from a bypass of one or both vessels. Additionally, we believe that asymptomatic patients with high scores may benefit from prophylactic treatment after accounting for other risk factors. However, these ideas need to be investigated and validated in further studies.

Interestingly, unlike the CMI diagnosis, the CSI-score showed a significant association with patient mortality. Although there was no statistically significant difference in mortality between patients with CMI and without, patients with a high score had significantly higher mortality than patients with a low score.

In addition, the score cannot predict the development of CMI symptoms in patients requiring CTA for another reason (e.g., peripheral arterial disease). In patients without suspected CMI, only the severity of visceral arterial stenosis can be calculated. Therefore, the new score cannot replace the clinical team that has the final decision on the diagnosis of CMI. However, the score complements the data obtained from the clinical examination.

This study is limited by its retrospective and single-center nature. Furthermore, since there is no validation cohort, the results might be biased due to overfitting issues. External validation of the score will be of great importance in evaluating the performance and prediction ability of the score. Additionally, the bootstrap analysis cannot replace an external validation. However, the advantage of the score is that it takes into account more information about the stenosis of every individual vessel. Therefore, we believe that it is a valuable and promising clinical instrument. Although a multidisciplinary consensus confirmed the diagnosis of CMI in most cases, misdiagnosis and delayed treatment can be expected in patients without CMI.

## 5. Conclusions

Patients with a higher CSI-score showed a strong association with the presence of chronic mesenteric ischemia. The new score can be used as an indicator of the risk of CMI and the need for prophylactic intervention in the future. However, the feasibility and validity of this score require further confirmation through future studies.

## Figures and Tables

**Figure 1 jcm-11-07420-f001:**
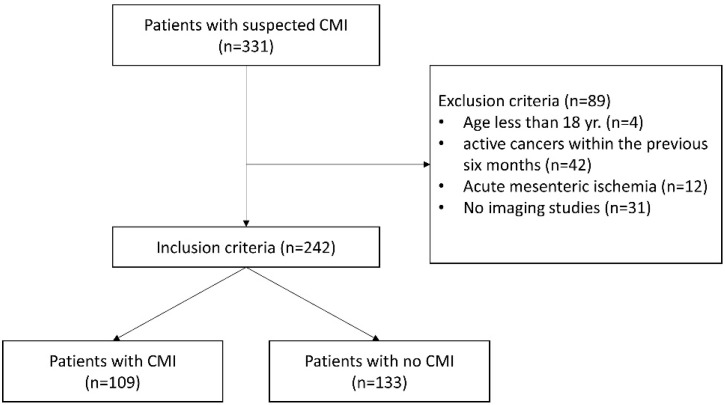
Flow chart showing patient inclusion and study screening process.

**Figure 2 jcm-11-07420-f002:**
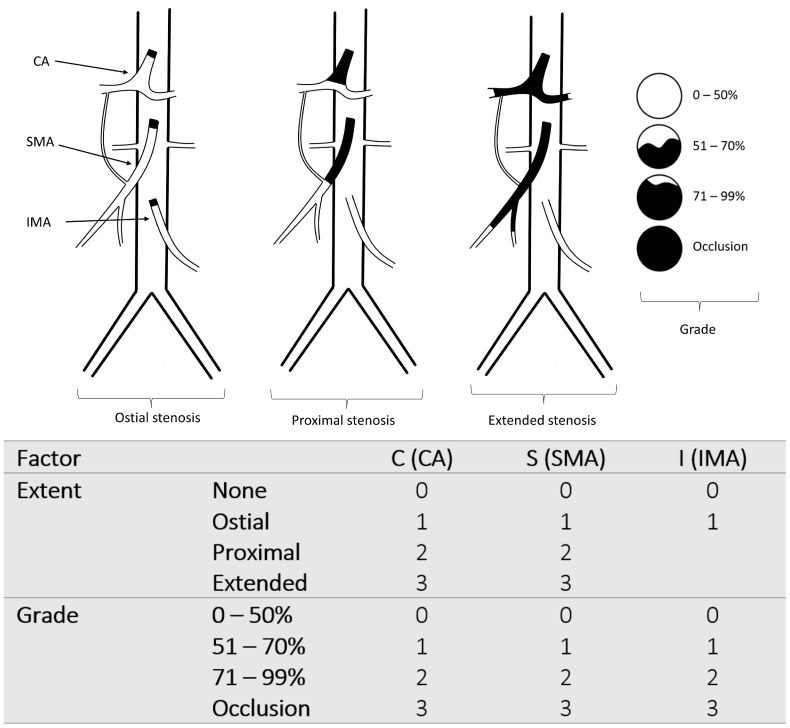
Classification and scoring model of the stenosis of the main visceral arteries. CA: celiac artery, SMA: superior mesenteric artery, IMA: inferior mesenteric artery.

**Figure 3 jcm-11-07420-f003:**
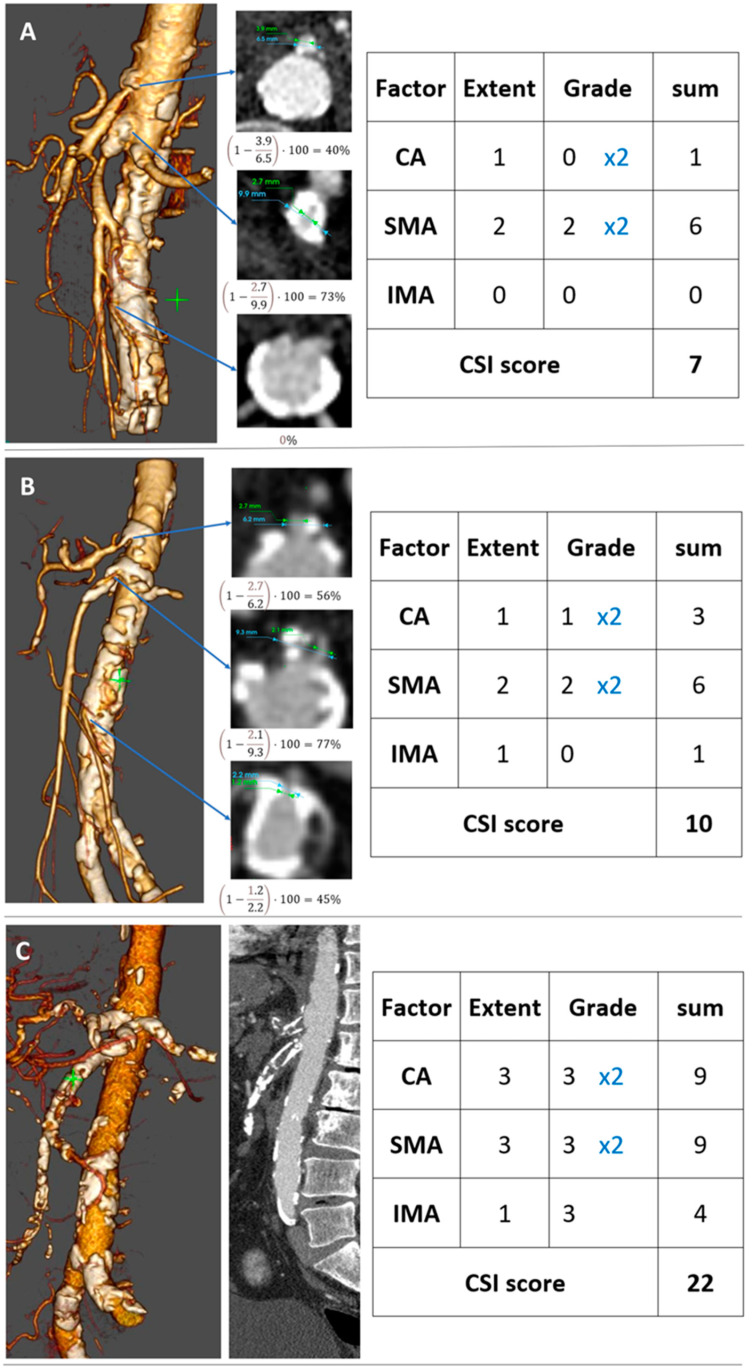
2D and 3D reconstructed CT angiography with sample calculations of the stenosis grade of the visceral arteries and the CSI-score for three patients. (**A**) Showing ostial (40%) stenosis of the CA, proximal (73%) of the SMA, with no stenosis of the IMA. (**B**) Showing ostial (56%) stenosis of the CA, proximal (77%) of the SMA, and ostial (45%) of the IMA. (**C**) Showing extended occlusion of the CA and SMA and ostial occlusion of the IMA.

**Figure 4 jcm-11-07420-f004:**
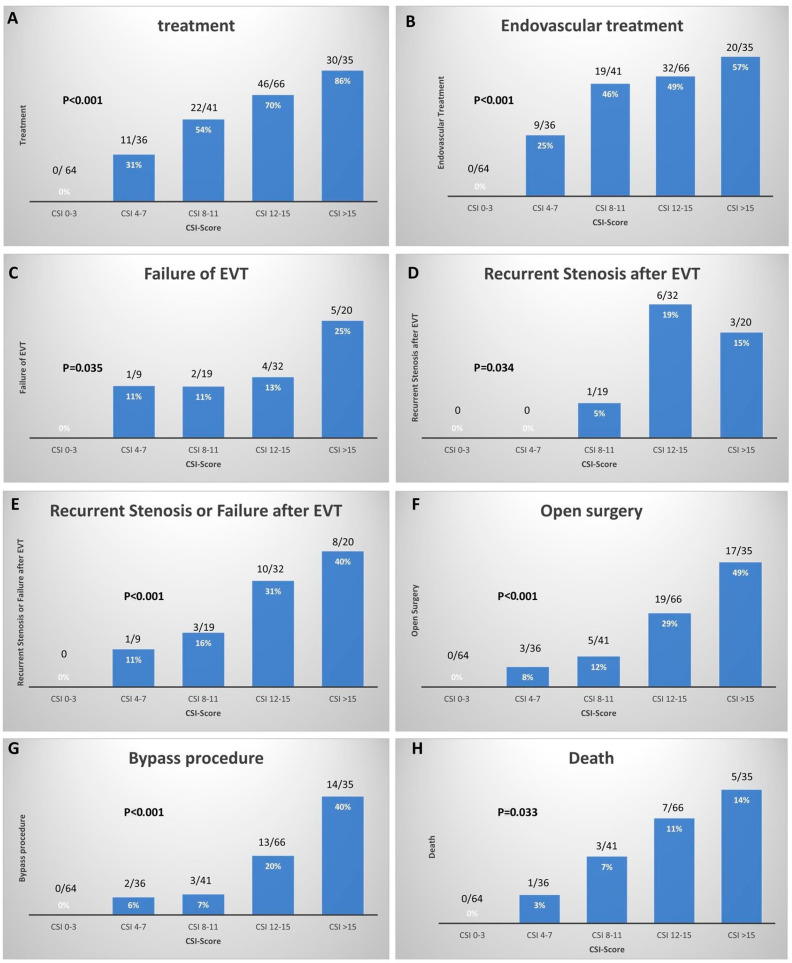
The histograms show the number and rates of patients in five CSI-score subgroups (0–3, 4–7, 8–11, 12–15, >15) stratified by treatment (**A**), endovascular treatment (EVT) (**B**), failure of EVT (**C**), recurrent stenosis after EVT (**D**), recurrent stenosis or failure after EVT (**E**), open surgery (**F**), bypass procedure (**G**), and death (**H**).

**Table 1 jcm-11-07420-t001:** Demographics and risk factors of the patients, number of affected vessels, and location of mesenteric artery stenosis.

Characteristics	Total	Low Score < 8	High Score ≥ 8	*p*
**Number**	242	100	142	
CMI	109 (45)	11 (11)	98 (69)	<0.001 *
**Demographics**				
Age (median, range, y)	71 (32–99)	71 (32–90)	71 (42–99)	0.926
Female	128 (53)	48 (48)	80 (56)	0.201
**Risk factors**				
Coronary artery disease	112 (46)	36 (36)	76 (54)	0.007 *
Diabetes mellitus	58 (24)	18 (18)	40 (28)	0.068
Hypertension	180 (74)	68 (68)	112 (79)	0.056
Hyperlipoproteinemia	64 (26)	23 (23)	41 (29)	0.308
Stroke	50 (21)	16 (16)	34 (24)	0.133
COPD	60 (25)	24 (24)	36 (25)	0.810
Chronic renal insufficiency	101 (42)	30 (30)	71 (50)	0.002 *
Smoking	75 (31)	28 (28)	47 (33)	0.398
Alcoholism	16 (7)	6 (6)	10 (7)	748
PAD	95 (39)	16 (16)	79 (57)	<0.001 *
**ASA class**				0.002 *
I	3 (1)	3 (3)	0	
II	54 (22)	31 (31)	23 (16)	
III	156 (66)	59 (59)	97 (68)	
IV	29 (12)	7 (7)	22 (16)	
**Number of affected vessels**				<0.001 *
None	46 (19)	46 (46)	0	
One	52 (22)	48 (48)	4 (3)	
Two	53 (22)	6 (6)	47 (33)	
Three	91 (38)	0	91 (64)	
**Stenotic arteries**				<0.001 *
None	46 (19)	46 (46)	0	
CA	22 (9)	20 (20)	2 (1)	
SMA	15 (6)	13 (13)	2 (1)	
IMA	15 (6)	15 (15)	0	
CA and SMA	34 (14)	2 (2)	32 (23)	
CA and IMA	6 (3)	2 (2)	4 (3)	
SMA and IMA	13 (5)	2 (2)	11 (8)	
CA and SMA and IMA	91 (38)	0	91 (64)	

CMI: chronic mesenteric ischemia, PAD: peripheral arterial disease, ASA class: American Society of Anesthesiologists Classification, *: *p*-value ≤ 0.05.

**Table 2 jcm-11-07420-t002:** Clinical presentation, gastrointestinal manifestations, and surgeries.

Characteristics	Total	Low Score < 8	High Score ≥ 8	*p*
**Number**	242	100	142	
**Clinical Presentation**				
Abdominal pain	147 (61)	27 (27)	120 (85)	<0.001 *
Postprandial pain	95 (39)	17 (17)	78 (55)	<0.001 *
Constant abdominal pain	52 (22)	10 (10)	42 (30)	<0.001 *
BMI	24 ± 5	25 ± 5	23 ± 5	0.009 *
Underweight BMI < 18.5	28 (12)	4 (4)	24 (17)	0.002 *
Weight loss	68 (28)	13 (13)	55 (38)	<0.001 *
5–10% of BM	33 (14)	5 (5)	28 (20)	0.001 *
>10%	35 (15)	8 (8)	27 (19)	0.016 *
Fear of food	10 (4)	1 (1)	9 (6)	0.050 *
Nausea	39 (16)	15 (15)	24 (17)	0.692
Vomiting	26 (11)	8 (8)	18 (13)	0.247
Loss of appetite	80 (33)	20 (20)	60 (42)	<0.001 *
Diarrhea	25 (10)	10 (10)	15 (11)	0.887
Exercise-induced abdominal pain	3 (1)	0	3 (2)	0.270
**Gastrointestinal manifestation**				
Ischemic gastritis	25 (10)	4 (4)	21 (15)	0.007 *
Colonic ischemia	44 (18)	19 (19)	25 (18)	0.782
Ischemic hepatitis	1(0.4)	0	1 (0.7)	
Upper GI bleeding	27 (11)	12 (12)	15 (11)	0.727
Lower GI bleeding	15 (6)	10 (10)	5 (4)	0.040 *
**Gastrointestinal surgery (all reasons)**	31 (13)	2 (2)	29 (20)	0.001 *
Colectomy	17 (7)	2 (2)	15 (11)	0.010 *
Small intestine resection	19 (8)	0	19 (13)	<0.001 *
Billroth II	3 (1)	0	3 (2)	0.270
**Invasive treatment**	109 (45)	11 (11)	98 (69)	<0.001 *
Primary EVT	80 (33)	9 (9)	71 (50)	<0.001 *
EVT primary success	69	8	61	
Failure of EVT	12	1	11	
Recurrent stenosis after EVT	10	0	10	
Failure or recurrence after EVT	22 (9)	1 (1)	21 (15)	<0.001 *
Open conversion after EVT	17 (7)	1 (1)	16 (11)	0.002 *
Primary open surgical treatment	44 (18)	3 (3)	41 (29)	<0.001 *
Bypass procedure	32 (13)	2 (2)	30 (21)	<0.001 *
**Outcome**				
Overall deaths	16 (7)	1 (1)	15 (11)	0.003*

BMI: body mass index (kg/m^2^), BM: body mass, EVT: endovascular treatment, CA: celiac artery, SMA: superior mesenteric artery, IMA: inferior mesenteric artery, GI: gastrointestinal, *: *p*-value ≤ 0.05.

## Data Availability

The data presented in this study are available on request from the corresponding author.

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
