# Peer review of "Development of a Novel Scoring Model to Estimate the Severity Grade of Mesenteric Artery Stenosis"

_jcm, 2022, doi:10.3390/jcm11247420_

Round 1
Reviewer 1 Report
- There is a need for a better guide to make clinical decisions, but this scoring system, in my opinion, does not contribute to that fact. It is primarily a descriptive study. The clinical relevance of the results are not enough reflected in the discussion/conclusion. The translation to the clinic (the final step in the scoring system) is missing. How to apply this scoring system in the clinic should be more extensively investigated and described.
- One of the discussion points, that patients with a higher score should possibly opt for bypass surgery instead of endovascular treatment, is an oversimplified conclusion. According to the European guidelines bypass surgery over endovascular recanalization is only chosen when patients are not eligible for endovascular recanalization or after failed endovascular recanalization. There are plenty patients with stenosis in two vessels that profit for a longer period from 1 or 2 stent placements.
- The higher the CSI score, the more need for treatment. This is supported with an AUS of 0.87, highlight this more in the conclusion and discussion.
- The IMA is also included in the scoring system, you might consider removing it from the scoring system.
- The novelty of the scoring system. There is some interesting and similar data published in Terlouw et al (2021) concerning this subject. Could you elaborate not referring to this article?
Author Response
Reviewer #1:
- There is a need for a better guide to make clinical decisions, but this scoring system, in my opinion, does not contribute to that fact. It is primarily a descriptive study. The clinical relevance of the results are not enough reflected in the discussion/conclusion. The translation to the clinic (the final step in the scoring system) is missing. How to apply this scoring system in the clinic should be more extensively investigated and described.
Response: We thank the reviewer for this important comment. CSI-score is independent of the clinical findings and is a pure morphological classification of the stenosis of the visceral arteries based on the CT angiography, as we described in the discussion. However, we found a strong association between this score and the clinical manifestations, treatment options, and outcomes.
“While the diagnosis of CMI is based on clinical findings and radiologic manifesta-tions [8, 9], the CSI-score is entirely independent of the clinical findings and is a pure morphological classification of the radiologic findings. However, we found a strong asso-ciation between this score and clinical manifestations, treatment options, and outcomes. The new scoring model demonstrated a high discriminatory ability between patients with CMI and those without. In addition, the score stratification showed that the percentage of patients treated with endovascular and open surgical methods, the recurrence or failure of the endovascular treatment, the need for a bypass procedure, and the mortality rates sig-nificantly increased in the subgroups. Based on these results, we not only need to classify the patients into CMI and non-CMI but also to correlate them with the score values. For example, treating CMI patients with a CSI-score of 5 differs from those with a CSI-score of 20 because we expect the degree of stenosis to be different.” (Redline page 12 lines 348-361)
According to the clinical presentation, we found that high-scored patients revealed more abdominal pain, lower body mass index, more weight loss, and loss of appetite.
“According to the clinical presentation, abdominal pain was the most frequent symp-tom in 147 (61%) patients and was more often in the high-scored patients (85% vs. 27%, p < .001). The classic postprandial pain symptoms were seen in 95 (39%) of the patients and were more often in the high-scored patients (55% vs. 17%, p < .001). Constant abdominal pain was the main symptom in 52 (22%) patients and was more often in the high-scored patients (30% vs. 10%, p < .001). The body mass index was lower in high-scored patients (23 vs. 25 kg/m2, p = .009). Additionally, the high-scored patients revealed more weight loss (38% vs. 13%, p < .001), and loss of appetite (42% vs. 20%, p < .001). Further diagnostic investigations, including gastroduodenoscopy, coloscopy, and CT-angiography, showed ischemic gastritis in 25 (10%) patients, ischemic hepatitis in one (0.4%) patient, and is-chemic colitis in 44 (18%) patients see Table 2.” (Redline page 8 lines 237-247)
Regarding the use of the score in the treatment and decision-making we added the following text to the discussion:
“According to the application of the score, we need the ultimate and detailed values of the CSI-score to make decisions. Detailed score values indicate the diseased narrowed blood vessels and the location, severity, and extent of the narrowing. For example, patients with ostial or proximal stenosis of the CA and SMA may benefit from endovascular treatment, and patients with extended stenosis or occlusion of the CA, SMA, or both may benefit from a bypass of one or both vessels. However, the application of the score needs to be investigated and validated in further studies.” (Redline page 14 lines 439-445).
- One of the discussion points, that patients with a higher score should possibly opt for bypass surgery instead of endovascular treatment, is an oversimplified conclusion. According to the European guidelines bypass surgery over endovascular recanalization is only chosen when patients are not eligible for endovascular recanalization or after failed endovascular recanalization. There are plenty patients with stenosis in two vessels that profit for a longer period from 1 or 2 stent placements.
Response: We agree with the reviewer. Endovascular treatment is the first option when possible due to its minimal invasiveness. However, none of the available studies provided clear criteria for choosing between endovascular and open options, and none considered the extension of stenosis in decision-making. In addition, it is well known from practical experience that the majority of the extended stenoses of the visceral arteries are not feasible for endovascular treatment, making it difficult to implant a stent-graft if possible and having poor results after balloon angioplasty. Therefore, it is reasonable to try endovascularly because of its lower risk than open surgery, especially in high-morbid patients, but as we revealed in our study, the recurrence and failure of the endovascular treatment and the need for a bypass procedure increased in parallel with the CSI-score. It is absolutely right that many patients with stenosis in two vessels profit for a longer period from 1 or 2 stent placements. However, what was the length and grade of stenosis in these patients?
For example:
Ostial (51-70%) stenosis of the CA and SMA results in a CSI-score of 6 (if IMA had no stenosis)
Ostial (71-99%) stenosis of the CA and SMA results in a CSI-score of 10
Extended occlusion of the CA and SMA results in a CSI-score of 18
Two vessel-stenosis doesn’t always mean a high score value.
- The higher the CSI score, the more need for treatment. This is supported with an AUS of 0.87, highlight this more in the conclusion and discussion.
Response: As suggested by the reviewer, we added this point to the conclusions and discussion.
- The IMA is also included in the scoring system, you might consider removing it from the scoring system.
Response: During the development of the score, we tested all possible modalities, including each single-vessel score, combining two vessels in one score, and three-vessel score. The best AUC value was obtained by using the following equation:
CSI-score=1*Cextent+2*Cgrade+1*Sextent+2*Sgrade+1*Iextent+1*Igrade
This equation shows that the grade of stenosis of the CA and SMA is more weighted than IMA. Additionally, we included only ostial stenosis of the IMA to avoid overestimating the role of the IMA.
Nevertheless, we don’t have to ignore that all three vessels have extensive collaterals, which are likely to have a compensating function in preventing ischemia after the occlusion of one or two vessels.
- The novelty of the scoring system. There is some interesting and similar data published in Terlouw et al (2021) concerning this subject. Could you elaborate not referring to this article?
Response: Many thanks for showing this important article. We regret that we inadvertently overlooked this important work. We started our research in 2020 and did not see the article of Terlouw et al. before. In addition, we used the term “chronic mesenteric ischemia” to search the literature, while the term “Chronic Gastrointestinal Ischemia” was used by Harki et al. We discussed these articles in detail and added these articles to our references.
“Harki et al. [8] developed a scoring model to predict the risk of chronic gastrointestinal is-chemia based on the radiological assessment of the mesenteric arteries and clinical char-acteristics. Only CA and SMA stenosis and the stenosis grade of <50%, 50% to 70%, or >70% were considered in this study. Other predictors for this scoring model included gender, weight loss, and cardiovascular disease. The results of this score were validated in other articles [9, 20]. In contrast to the present study, none of the existing studies reported on the extension of stenosis of the visceral artery as a predictor of CMI and outcome. It is well known that the extension of the stenosis beyond the collateral arteries affects the compensation of the celiac mesenteric arterial system [21, 22].” (Redline page 13 lines 373-378).
Reviewer 2 Report
Dear Editor,
I have read the paper by Safwan Omran et al with interest. The authors propose a new scoring model to estimate the severity grade of the stenosis of the mesenteric arteries and its relationship with the development of chronic mesenteric ischemia and to guide the treatment of CMI patients
The Study found that the new scoring model can 40 estimate the severity grade of the stenosis of the mesenteric arteries and can be helpful to predict 41 the existence of symptoms and guide the treatment of patients with chronic mesenteric ischemia.
The topic of the paper fits within the scope of JCM and the novelty is present. The paper is well written, easy to read and can bring new information in scientific community
Minor revision:
1. The authors must be consistent with the verb tenses used in the manuscript.
2. In the subsection “The new classification and endpoints”, please state what are the endpoints?
3. The Discussion section needs to be revised. There is too much information that is already stated in the Results and Methods sections. Shorten the information and improve the quality of the research with comparing the results with other articles.
4. Add one more phrase in the Conclusion section.
5. Please correct the following:
Change “cutoff” to “cut-off”
Line 131: (see figures 2) -> (see figure 2)
Lines 131-132: Please reformulate „CA is abbreviated by C, SMA by 131 (S), and IMA by (I).” TO -> „CA is abbreviated by „C”, SMA by „S”, and IMA by „I”.
… etc. Corrections need to be done all over the manuscript.
6. Please place all the figures and tables right where they are mentioned in the text.
7. For Figure 2:
Please separate the figure and the table, and use a table made in Word, with the font of size 10.
8. For Table 1:
There is no title for the table? Please revise.
Add all abreviations below the table.
Make sure all data in the table is liniar and at the right place.
9. For Table 2:
Add all abreviations below the table.
Make sure all data in the table is liniar and at the right place.
10. For References:
The references need to be in accordance with the instructions in the template. The list should be with 1., 2., 3., 4. … etc. Also, add the DOI for each reference.
Please see the Authors’ Instructions: “In the text, reference numbers should be placed in square brackets [ ], and placed before the punctuation; for example [1], [1–3] or [1,3]. For embedded citations in the text with pagination, use both parentheses and brackets to indicate the reference number and page numbers; for example [5] (p. 10). or [6] (pp. 101–105). The reference list should include the full title, as recommended by the ACS style guide. Style files for Endnote and Zotero are available.”
I want to congratulate the authors for the paper.
Kind regards.
Author Response
Dear Editor,
I have read the paper by Safwan Omran et al with interest. The authors propose a new scoring model to estimate the severity grade of the stenosis of the mesenteric arteries and its relationship with the development of chronic mesenteric ischemia and to guide the treatment of CMI patients
The Study found that the new scoring model can estimate the severity grade of the stenosis of the mesenteric arteries and can be helpful to predict the existence of symptoms and guide the treatment of patients with chronic mesenteric ischemia.
The topic of the paper fits within the scope of JCM and the novelty is present. The paper is well written, easy to read and can bring new information in scientific community.
Response: We thank the reviewer and appreciate his positive comment.
Minor revision:
- The authors must be consistent with the verb tenses used in the manuscript.
Response: As suggested by the reviewer, we made a language revision of the entire text.
- In the subsection “The new classification and endpoints”, please state what are the endpoints?
Response: As suggested by the reviewer, we added the primary and secondary endpoints to the methods.
“The primary endpoint was the association of CSI-score with chronic mesenteric ischemia. Secondary endpoints were mortality, the need for treatment, endovascular failure or recurrence, and the need for bypass procedures.” (Redline page 5 lines 165-167)
- The Discussion section needs to be revised. There is too much information that is already stated in the Results and Methods sections. Shorten the information and improve the quality of the research with comparing the results with other articles.
Response: As suggested by the reviewer, we made a revision of the discussion section.
- Add one more phrase in the Conclusion section.
Response: As suggested, we added one more phrase to the conclusions.
- Please correct the following:
Change “cutoff” to “cut-off”
Line 131: (see figures 2) -> (see figure 2)
Lines 131-132: Please reformulate „CA is abbreviated by C, SMA by 131 (S), and IMA by (I).” TO -> „CA is abbreviated by „C”, SMA by „S”, and IMA by „I”.
… etc. Corrections need to be done all over the manuscript.
Response: As suggested, we made the corrections.
- Please place all the figures and tables right where they are mentioned in the text.
Response: We placed the figures and tables in the right place.
- For Figure 2:
Please separate the figure and the table, and use a table made in Word, with the font of size 10.
Response: As suggested, we separated the figure and the table.
- For Table 1:
There is no title for the table? Please revise.
Add all abreviations below the table.
Make sure all data in the table is liniar and at the right place.
Response: We made the suggested corrections.
- For Table 2:
Add all abreviations below the table.
Make sure all data in the table is liniar and at the right place.
Response: We made the suggested corrections.
- For References:
The references need to be in accordance with the instructions in the template. The list should be with 1., 2., 3., 4. … etc. Also, add the DOI for each reference.
Please see the Authors’ Instructions: “In the text, reference numbers should be placed in square brackets [ ], and placed before the punctuation; for example [1], [1–3] or [1,3]. For embedded citations in the text with pagination, use both parentheses and brackets to indicate the reference number and page numbers; for example [5] (p. 10). or [6] (pp. 101–105). The reference list should include the full title, as recommended by the ACS style guide. Style files for Endnote and Zotero are available.”
Response: We added the DOI to the references.
I want to congratulate the authors for the paper.
Response: We really appreciate this positive feedback.